# Regulatory scripting: Stakeholder participation in food and drug administration medical device advisory meetings

**Shelley K. White**[1]☺*, **Valerie Leiter**[1]☺, **Mi H. Le**[2]‡, **Caitlyn K. Helms**[1]‡

**1** Departments of Public Health and Sociology, Simmons University, Boston, Massachusetts, United States of America, **2** Departments of Biostatistics and Epidemiology, Boston University School of Public Health, Boston, Massachusetts, United States of America

☺ These authors contributed equally to this work.
‡ MHL and CKH also contributed equally to this work.
* shelley.white@simmons.edu

**Data Availability Statement:** The data relevant to this study are available from OSF at https://osf.io/w2ced/.

## Abstract

In an age of biomedicalization, medical devices have become more common and more technologically complicated, and adverse events associated with medical devices have increased. The U.S. Food and Drug Administration (FDA) relies on advisory panels to assist in regulatory decision making regarding medical devices. Public meetings held by these advisory panels allow stakeholders to testify, presenting evidence and recommendations, according to careful procedural standards. This research examines the participation of six stakeholder groups (patients, advocates, physicians, researchers, industry representatives and FDA representatives) in FDA panel meetings focused on the safety of implantable medical devices between 2010–2020. We use qualitative and quantitative methods to analyze speakers' opportunities for participation, bases of evidence, and recommendations, applying the concept of 'scripting' to understand how this participation is shaped by regulatory structures. Regression analysis demonstrates statistically significant differences in speaking time, where researchers, industry, and FDA representatives had longer opening remarks and more exchanges with FDA panelists than patients. Patients, advocates and physicians shared the least amount of speaking time, and were the parties most likely to leverage patients' embodied knowledge and recommend the most stringent regulatory actions like recalls. Meanwhile, researchers, FDA, and industry representatives rely on scientific evidence and, with physicians, recommend actions that preserve medical technology access and clinical autonomy. This research highlights the scripted nature of public participation and the types of knowledge considered in medical device policymaking.

## Introduction

Across its regulatory activities, the U.S. Food and Drug Administration (FDA) relies on advisory committees to provide independent, expert advice. The FDA's overall Medical Devices Advisory Committee has 18 panels, each of which focuses on different topic areas, including general and plastic surgery devices, immunology devices, and obstetrics and gynecological

**Funding:** The authors received internal funding from the Simmons University Undergraduate Faculty-Student Collaborative Fellowship to hire student research assistantships in support of this study (who became the third and fourth authors of this manuscript). The funders had no role in study design, data collection and analysis, decision to publish, or preparation of the manuscript.

**Competing interests:** No authors have competing interests.

devices. These advisory structures are essential to the FDA's mission of protecting and advancing public health [2]. Per the FDA, 'the primary role of the FDA advisory committee is to: provide independent expert advice to the Agency in its evaluation of these regulated products, and help the agency move toward making sound decisions based upon reasonable application of sound scientific principles' [1]. Advisory committee meetings also provide a forum for public voices, as 'the FDA uses advisory committees to. . .encourage patients, health care providers, and other interested people to share their views during the open public hearing or by submitting comments on the open docket' [1]. Every advisory meeting includes an Open Public Hearing (OPH) to allow for public participation in the policy process. Various stakeholders provide testimony according to careful procedural standards dictated by FDA policy and mediated by committee or panel chairs [2, 3]. As outlined below, OPHs are a key mechanism through which public stakeholders can have input into the FDA's medical device regulatory system.

This article examines the dynamics of participation within meetings of the FDA's 18 medical device advisory panels. Medical devices represent a growing segment of global healthcare, where the U.S. has the largest market with a projected value of $208 billion by 2023 and over 6,500 medical device companies [4]. Nonetheless, medical devices receive far less attention than pharmaceuticals, prompting Faulkner to designate them as the 'Cinderella' of medical technologies [5]. As medical device use has grown, so have device-related adverse events, triggering the Institute of Medicine (IOM) and the U.S. Government Accountability Office (GAO) to review safety concerns with FDA medical device regulation [6, 7]. Medical device safety is a central concern of FDA panel meetings, whether they are scheduled for pre-market reviews or in response to post-market safety issues.

Prior studies have considered the dynamics of participation and voting behavior of medical device advisory committee members [8]. Here, we consider the dynamics of participation of six stakeholder groups (patients, advocates, physicians, researchers, industry representatives, and FDA representatives) during the presentation and question and answer sections of meetings. We examine medical device panel meetings between 2010–2020 focused on post-market device safety concerns, analyzing and comparing how participation varied by type of stakeholder. We analyze the amount of speaking time (opportunities to make statements and engage in exchanges with FDA panelists) these stakeholders had during the advisory meetings, the types of evidence they relied upon in their testimonies, and the types of recommendations they advanced. This study provides insight on what it means for there to be public participation in health policy-making in the context of the FDA's regulation of, and response to safety concerns with, medical devices.

## FDA medical device regulation

The FDA regulates three classes of medical devices based on intended use and risk as established by the 1976 Medical Devices Act. Class I are low-risk, non-invasive devices. Class II are moderate-risk devices that may include surgical and implantable products. Class III are high-risk devices, including life-saving technologies. Class II and III devices may be subject to device panel meetings, the focus of this research. The FDA has distinct pre-market requirements for these device classes. It typically 'approves' Class III devices through a Premarket Approval (PMA) process, requiring that manufacturers provide clinical trial evidence on safety and efficacy [3, 8]. In contrast, it 'clears' Class II devices for the market through a '510(k) process,' requiring manufacturers to notify the FDA of intent to market, and to demonstrate that the device is 'substantially equivalent' to a 'predicate,' or legally-marketed device [8–10]. Only 1% of US medical devices have been approved through the Class III PMA process, leading

some to criticize the FDA for 'regulatory reticence' [11]. Importantly, the public may not perceive the differences of device 'approval' versus device 'clearance,' and may assume that FDA device regulation is equivalent to the FDA's more stringent pharmaceutical regulation [12].

The FDA tracks adverse event reports that are associated with marketed medical devices. Such reports increased 15% per year between 2001–2009, then doubled between 2008 and 2011, reaching 400,000 in 2011 [13, 14]. Implantable devices have become more common in medical care, and more technologically complicated. Adverse events include immediate issues like post-surgical infections as well as latent events that can take years to evidence, such as hip implants shedding metal particles, surgical mesh eroding tissues, or female sterilization devices spurring chronic inflammatory disorders. Adverse events associated with implantable devices are particularly complicated because the devices are meant to be implanted permanently, patients cannot remove them without medical assistance, and they sometimes require revision surgeries [15]. Therefore, we chose to focus our study on the higher-risk group of implantable devices.

## FDA medical device advisory meetings

The concept of scripting is helpful in considering the structure of FDA advisory meetings. In actor-network theory, scripting reveals how technical objects set forth a 'framework of action' for those interacting with the object, based on the vision pre-scribed by the object's inventor. Actors may or may not conform to the object's embedded script [16]. Timmermans and Berg examine how medical protocols act as technoscientific scripts, which are negotiated dynamically by the medical personnel implementing them and even the patients subject to them [17]. Protocols, such as those guiding cardiac resuscitation, provide a script for a sequence of expected actions on the part of the medical team. In these medical settings, key actors, particularly physicians, make decisions about when, how, and to what extent to apply protocols. In the present study, we consider how the FDA's protocol for the OPH creates a specific script which predetermines the extent to which different actors' testimony and evidence is given space for consideration. While the script is negotiated somewhat dynamically, it is largely the advisory panel members who are empowered to shape the script through their use of the 'question and answer' periods after each group of speakers.

FDA advisory meetings have served as the longest-standing mechanism for public input into medical device regulatory proceedings. The structure of FDA advisory meetings is dictated by Title 21, Part 14 of the Code of Federal Regulations, and presented through FDA public guidelines [2]. Committees and panels must allow at least one hour for the OPH portion of a meeting, and interested speakers must apply by FDA deadlines. Speakers are typically allotted 5–10 minutes for testimonies; if there are too many interested speakers, the FDA may reduce allotted times per speaker, ask speakers to combine testimonies, use a lottery system, or extend the OPH. Speakers are given timed warnings and microphones are silenced when their time expires [2]. OPH speakers may include patients, physicians and other interested public parties. Industry representatives, as sponsors of the technology, are treated differently under Section 513 of the Food, Drug and Cosmetic Act. Along with FDA speakers, they are typically allotted 60 to 90 minutes [18]. Every FDA advisory meeting adheres to this overall structure. However, across all guidelines, FDA-appointed committee and panel chairs are given discretion to manage speaker presentations and meeting schedules. Additionally, question and answer periods are incorporated into advisory committee meetings; after each panel of speakers, the FDA panel may ask speakers questions or make requests for clarification.

## Dynamics of policy participation

Analyzing participation in policy-making processes can highlight power dynamics and differentials that shape stakeholders' voices, particularly those of patients. Within health-based social movements, patients generally rely on embodied knowledge, and also build lay expertise to advance 'evidence-based activism' [19–22]. History provides powerful examples of lay individuals emerging as 'active patients' leveraging their 'local knowledge' to challenge scientific authority [23–25]. Citizen participation that aims to shape health policy takes several forms and involves different uses of evidence depending on the circumstances and targets [26]. Yet public and patient involvement across health and health policy settings is often conflated into '*the* patient perspective,' with an assumed singularity [27].

Increased channels for citizen participation in regulatory spaces has become an 'international imperative,' and implies reliance on a variety of expert and non-expert inputs [27]. However, while 'sustained experience' and engagement around a particular issue could qualify both citizens and scientists as experts, citizens are often quickly discounted and thus rely on both contention and cooperation [28, 29]. Women's voices, in particular, disappear within regulatory and legal spaces as their personal stories become subordinated to the 'statistical victim,' considered more influential than 'emotional' accounts [30, 31]. Across these contexts, questions arise regarding the extent to which patients' experiential knowledge is valued as testimony, particularly when weighed against other types of knowledge and evidence [32].

Studies of regulatory decision-making also examine the participation of physicians and other scientific experts, industry representatives, and regulatory authorities. Jasanoff examines how scientists' authority can be jeopardized as the indeterminacy of science is exposed, and science itself is deconstructed and reconstructed at the hands of regulators [33]. Industry often pushes for a separation of science and policy by attacking 'bad science' and regulatory agencies' competence. Each party has a trajectory that informs its position. Scientists aim to maintain the prestige and authority they have honed through a history of professionalization processes, and to maintain confidence in their objectivity. Physicians are reluctant to relinquish their autonomy in determining appropriate technologies and practices [34–36]. Meanwhile, industry functions from a profit motive, often leveraging scientific knowledge toward liberalizing regulation of its products [37, 38].

Within FDA advisory committees, there are asymmetrical incentives for participation of different types of stakeholders, whereby industry groups will have more at stake and thereby greater participation than average citizens, leading to 'regulatory capture' [39]. Industry is 'permitted privileged strategic access to, and involvement with, government regulatory policy over and above any other interest group' [40]. Recent studies also demonstrate industry's indirect influence, where 30% or more of public speakers at FDA advisory committee meetings have been sponsored by industry (e.g. reimbursement for travel and expenses to attend the meeting) or have had undisclosed conflicts of interest [41, 42]. Finally, policy-makers serve as the arbiter of risk [43], determining who is qualified to serve as 'expert' in these policy spaces [31, 44].

In examining FDA medical device advisory meetings in this study, we ask two inter-related research questions: How is speaking time distributed across types of stakeholders during FDA advisory meetings, and how is this influenced by the panels' discretionary use of the question and answer period? Which types of evidence and recommendations do stakeholders leverage in their testimonies?

## Data and methods

This study relies on a multistage sampling plan for data collection and analysis. All data were accessed through the FDA's publicly available databases and archives, and therefore this

project was exempt from Institutional Review Board review. We took a mixed methods approach combining both qualitative and quantitative methods [45, 46]. For quantitative analysis, we used SPSS to run frequency distributions, crosstabulations with chi-square statistics, analysis of variance (for data exploration prior to modeling only; results not reported), and ordinary least squares (OLS) regression. For qualitative analysis, we used Nvivo 12 software to engage in thematic content analysis of meeting transcripts, combining deductive and inductive approaches to coding, and using cross-checks by multiple coders for inter-coder reliability [47, 48].

## Meeting-level sampling

The FDA website archives the meetings of all 18 panels of the Medical Devices Advisory Committee, the source of our data collection [49]. Meeting summaries are available from 2000 onward, marking the start of our sampling. Data for all 18 panels were collected from 2000–2020, resulting in a sample of 366 meetings. For each FDA panel meeting, the FDA archive provides, minimally, a summary of several paragraphs identifying the device of focus, the meeting goals, and an overview of the proceedings. Based on these FDA-authored summaries, and any meeting materials provided, each meeting was coded by descriptors of the device and by meeting purpose.

For meeting purpose, we inductively identified 13 reasons why panel meetings occurred based on FDA guidance regarding advisory committee purposes (see Table 1) [1, 3]. The majority (58%) focused on device approvals and clearances, as well as expanded uses of marketed devices, with 44% devoted to initial Class III PMA approvals. Overall, 165 of the meetings (45%) focused on implantable devices, and of these, 72% of meetings were pre-market focused, including 57% on PMA approvals. Post-market safety concerns represented 24% of all meetings and 22% of meetings on implantable devices.

From the 366 total meetings, we focused our analysis on the 165 meetings concerned with implantable devices in which the primary purpose was to discuss device safety (35 total). This sample was further narrowed to the 19 meetings for which all forms of documentation

**Table 1. FDA medical device meetings purpose, 2000–2020.**

| Meeting Purpose | Total Percentage (N = 366) | Implantable Percentage (N = 165) |
|---|---|---|
| *Device Approvals and Clearances* | | |
| • PMA Approval | 43% | 57% |
| • PMA Supplement | 8% | 11% |
| • 510(k) Clearance | 4% | 1% |
| • 510(k) Reevaluation | 1% | 1% |
| • Humanitarian Device Exemption | 1% | 2% |
| • De Novo Application | 1% | 0% |
| *Classification Considerations* | | |
| • Initial Classification of Pre-Amendment Devices | 9% | 3% |
| • Potential Down-Reclassification (lower risk) | 5% | 1% |
| • Class III Pre-amendment devices in Class II | 4% | 2% |
| *Safety Related* | | |
| • Device Safety Review | 11% | 9% |
| • Clinical Study Design Review | 9% | 10% |
| • FDA Draft Guidance | 2% | 2% |
| • Potential Up-Reclassification (higher risk) | 2% | 1% |

(Executive and/or 24-Hour Summary, Agenda, and Meeting Transcripts) were archived by the FDA and available for analysis. Twelve were one-day meetings and 7 were two-day meetings. Transcripts ranged from 217 to 765 pages, resulting in 7,219 transcript pages of data that were analyzed.

## Quantitative speaker-level sampling and analysis

The 19 meetings coded at the speaker level took place between 2010 and 2020 and included 9 of the 18 medical device panels (5 Circulatory System, 4 Neurological, 3 OB/GYN, 2 General and Plastic Surgery, and one each of: Dental; Ear, Nose and Throat; General Hospital; Immunology; and Orthopedic and Rehabilitation). Five of these meetings focused on women's devices (breast implant meetings in 2011 and 2019, surgical mesh meetings in 2011 and 2019, and a 2015 meeting on the Essure sterilization device), and the remainder were non-sex-specific (such as metal-on-metal hips, dental amalgam, and cochlear implants).

For these 19 meetings, we created a speaker-level database that recorded each contributing speaker (789 total). A contributing speaker refers to any speaker listed in the meeting agenda of the transcript, excluding press contacts and FDA panel members. Each speaker was coded by type of stakeholder as follows: patients (also including spouses or family members speaking on behalf of patients); advocates (representing a formal organization or collective, like patient support groups); solo physicians (speaking individually, from their own clinical practice experience); professional organizational physicians (representing professional organizations like the American Dental Association); researchers (including material scientists and clinical researchers); industry representatives (typically CEOs and other executives of device companies but also including physicians, researchers, patients, and advocates sponsored by industry, aligned with prior research on the topic [41, 42]); and FDA representatives (typically staff members not represented on the panel). All speakers were coded based on speaker titles listed in meeting agendas and speakers' self-introductions and disclosures of conflicts of interest, and the categories of speakers were mutually exclusive. This coding was performed with confirmatory cross-checks among all researchers.

Transcripts were coded initially to analyze the speaking time and discussion time that speakers were granted to contribute during the meeting, measured in three ways, aligned with prior research examining FDA panel proceedings [8]: (1) word count captured within a speaker's primary substantive presentation; (2) number of exchanges between a speaker and a panelist during question and answer sections of meetings (which occur when a panelist initiates further dialogue with a speaker, typically through a question or clarifying statement); and (3) word count from the speaker that resulted from any exchange initiated by a panelist. Word counts were obtained using Microsoft Word. Exchanges were recorded in a database that noted the panelist who started the exchange, the speaker addressed, the number of exchanges between the pair, and any additional word count. Speaking time was then analyzed quantitatively by type of stakeholder.

## Qualitative speaker-level sampling and analysis

From the 19 meetings coded for speaking time, we sampled a subset of meetings in which patients represented 10% or more of the total speaker population. The 9 meetings that included at least 10% patient speakers were held by 6 of the 18 medical device panels (3 OB/ GYN; 2 General and Plastic Surgery; and one each of Dental; Immunology; Neurological; and Orthopedic and Rehabilitation), and included all five women's devices. We purposely sampled meetings by patient participation to ensure sufficient patient representation for

comparative analysis by speaker type. Six of these 9 were two-day meetings, resulting in 4,362 pages of transcript data.

Transcripts were analyzed using NVivo 12 software. An initial codebook was established to identify categories of evidence and recommendations advanced by speakers. The codebook was informed deductively by our understanding of FDA policy and available actions, and created inductively through initial review of a sample of transcripts by all coders to establish inclusion and exclusion criteria [47, 48]. Following initial inter-rater alignment, line-by-line coding was conducted by one coder, with cross-checks by at least one other coder. Each speaker's presentation was coded as using any of the 11 bodies of evidence that they used. Three categories stemmed from direct experience: personal patient experience (speaker offers personal, embodied experience); other patients' experience (speaker discusses the personal experiences of others); and personal clinical experience (speaker discusses their direct clinical experience, distinguished from secondary data). The remaining categories were data-driven: literature review, material science, clinical trials data, MAUDE (the FDA's adverse events reporting database), alternative registries (device registries from other countries, or U.S.-based private registries), post-market follow up studies, and collected own data (where an individual or organization has collected its own, typically unpublished, data). Finally, a category of other evidence was included, capturing evidence that did not reflect the prior categories.

Each speaker's presentation was also categorized into one or more of the 10 types of recommendations that we identified through our coding: no recommended changes; informed consent (create or improve formal procedures for patients); improved labeling or patient communication (more general references to patient education and healthcare messaging, including from the FDA directly to the public); mandatory registry (require that all implanted medical devices are registered to allow for monitoring and follow-up); physician training (add or improve physician training in device-related surgical practices, whether required or voluntary); pre-market clinical trials (require increased safety review for new devices); recall (remove the device from market use); reclassify (change the device from Class II to Class III, requiring more stringent regulation); post-market studies (to be ordered by the FDA to monitor for emergent safety issues); and other (capturing additional recommendations). Some speakers made no recommendations.

## Results

### Speaking time at FDA meetings

Table 2 presents the distribution of speakers. Industry speakers made up 30% of speakers across all meetings; these speakers were primarily device industry corporate representatives, but also included representatives of adjacent industries such as blood testing and medical software manufacturers, as well as patients, physicians, and researchers sponsored by industry. FDA representatives were the next largest group (21%), followed by patients (16%), advocates (11%), physicians representing professional organizations (10%), and researchers and solo physicians (6% each). Here we see that industry and FDA speakers made up just over half of all speakers combined. At meetings concerned with women's medical devices (5 of 19 meetings), there were greater proportions of patients (24%) and advocates (14%), with lower proportions of industry (23%) and FDA representatives (16%).

Table 3 analyzes participants' speaking time. In all of the regression models, patients were the omitted, comparison category, to keep the focus on their voices relative to the voices of other types of participants. There are two models for each of the three outcome measures. The first model for each simply regresses the dependent variable on the types of speakers. The second model enters a dummy variable that measures if the hearing was about a women's health

**Table 2. Speakers for all device and women's device meetings concerned with post-market safety.**

| Type of Speaker | Total Percentage | Women's Device Percentage |
|---|---|---|
| | (N = 789) | (N = 388) |
| Industry (total): | 30% | 23% |
| • Corporate representatives | 12% | 13% |
| • Physician | 10% | 6% |
| • Adjacent industries | 2% | 2% |
| • Patient | 2% | 1% |
| • Researcher | 4% | 1% |
| FDA Representative | 21% | 16% |
| Patient (Individual) | 16% | 24% |
| Advocate | 11% | 14% |
| Physician (Professional Organization) | 10% | 11% |
| Physician (Solo) | 6% | 8% |
| Researcher | 6% | 4% |

device. That variable was controlled for here because there were a large number of speakers for the women's health device meetings, and as a result, speakers had a shorter amount of time in which to make their statements. By controlling for that, we ensured that any differences between patient speaking time and other participants' speaking time are not an artifact of the shorter statements that patients got to make during the women's health device meetings.

In terms of initial statement word count, Table 3 shows that speakers in women's health device meetings shared 163.83 fewer words, on average, but there were no differences for the number of exchanges or the exchange word counts. Compared with patients, on average researchers uttered 686.46 more introductory words, FDA representatives uttered 486.02 more

**Table 3. Regression estimates of introductory word count, exchange count, and exchange word count (N = 789).**

| | Introductory Word Count | Introductory Word Count | Exchange Count | Exchange Count | Exchange Word Count | Exchange Word Count |
|---|---|---|---|---|---|---|
| | Model 1 | Model 2 | Model 1 | Model 2 | Model 1 | Model 2 |
| Advocate | 187.35 | 169.57 | -0.244 | -0.254 | -22.76 | -26.05 |
| | (124.48) | (124.29) | (0.99) | (0.99) | (85.87) | (86.05) |
| FDA Rep | 538.16*** | 486.02*** | 5.00*** | 4.97*** | 355.78*** | 345.64*** |
| | (105.38) | (107.14) | (0.84) | (0.86) | (72.69) | (74.18) |
| Industry | 466.77*** | 415.74*** | 3.61*** | 3.58*** | 373.32*** | 363.88*** |
| | (97.98) | (99.84) | (0.78) | (0.80) | (67.59) | (69.12) |
| Physician (Prof. Org.) | 269.06* | 240.51 | 1.15 | 1.15 | 140.91 | 135.63 |
| | (126.77) | (126.89) | (1.01) | (1.02) | (87.44) | (87.84) |
| Physician (Solo) | 150.85 | 142.25 | -.359 | -0.36 | -27.81 | -29.40 |
| | (156.11) | (155.65) | (1.25) | (1.25) | (107.69) | (107.75) |
| Researcher | 746.90*** | 686.46*** | 3.86** | 3.82* | 288.20** | 277.02** |
| | (147.91) | (149.46) | (1.18) | (1.20) | (102.03) | (103.47) |
| Women's health | | -163.83* | | -0.94 | | -30.29 |
| | | (66.46) | | (0.53) | | (46.01) |

* p < .05

** p < .01

*** p < .001

words, and industry representatives uttered 415.72 more words. In terms of the number of exchanges with panelists, FDA representatives had 4.97 more, researchers had 3.82 more, and industry representatives had 3.58 more exchanges than patients. Turning to the words uttered in the exchanges with panelists, industry representatives uttered 363.88 more words, FDA representatives uttered 345.64 more, and researchers uttered 277.02 more words than patients. There were no statistically significant differences in speaking time between patients and advocates, solo physicians, or physicians who represented professional organizations.

## Speaker evidence and recommendations

In terms of evidence leveraged by speakers, there were statistically significant differences in the use of clinical trials, own data, literature reviews, material science, personal patient's experiences, other patient's experiences, personal clinical experience, and post-market evidence (see Table 4). Physicians representing professional organizations (21%), advocates (16%) and solo physicians and researchers (14% each) were most likely to cite clinical trials as evidence. Researchers (26%), industry (12%) and solo physicians (11%) were most likely to cite data that they had collected themselves. Advocates (42%), researchers (41%), solo physicians (36%), and physicians representing a professional organization (31%) were most likely to cite evidence from a medical literature review, and this was the most common source leveraged by FDA representatives (18%). Researchers (30%) were much more likely than other speakers to cite material science as evidence; many researchers were invited specifically to provide that expertise. Advocates were most likely to cite evidence from post-market studies (15%).

Patients were most likely to cite their own patient experiences (84%), followed by advocates (19%). Advocates (30%) and patients (22%) were most likely to cite other patients' experiences,

**Table 4. Types of evidence by speaker type (N = 789)[1].**

| | Advocate (N = 86) | FDA Rep (N = 163) | Industry (N = 235) | Patient (N = 129) | Physician, Professional Organization (N = 81) | Physician, Solo (N = 44) | Researcher (N = 51) |
|---|---|---|---|---|---|---|---|
| Alternative registries | 1% | 4% | 3% | 2% | 7% | 2% | 6% |
| Clinical Trials*** | 16% | 7% | 12% | 2% | 21% | 14% | 14% |
| Own Data*** | 7% | 4% | 12% | 2% | 5% | 11% | 26% |
| Literature Review*** | 42% | 18% | 20% | 12% | 31% | 36% | 41% |
| Material Science*** | 9% | 7% | 7% | 2% | 5% | 2% | 30% |
| MAUDE | 9% | 7% | 3% | 2% | 4% | 4% | 6% |
| Post-market evidence** | 15% | 6% | 6% | 2% | 5% | 4% | 0% |
| Personal patient experience*** | 19% | 0% | 3% | 84% | 11% | 9% | 0% |
| Other patient experience*** | 30% | 1% | 3% | 22% | 4% | 14% | 2% |
| Personal clinical experience*** | 7% | 3% | 14% | 8% | 32% | 68% | 18% |
| Other evidence | 4% | 1% | 0% | 0% | 2% | 2% | 0% |
| No specific evidence cited*** | 22% | 76% | 65% | 7% | 32% | 11% | 23% |

* $p < .05$

** $p < .01$

*** $p < .001$

[1]Note: Speakers often leveraged more than one category of evidence.

followed by solo physicians (14%). Solo physicians relied heavily on their own clinical experiences (68%), followed by physicians representing professional organizations (32%), and industry speakers (14%). Very few speakers used other forms of evidence, such as reports from the World Health Organization or data from unpublished university research.

Here we share quotes that exemplify the different evidence bases that speakers leveraged. Patients' experiential evidence was presented in more of an emotional tone than the data-driven testimonies:

> 'The MoM hip was removed. The surgeon said he was astonished to see such a mess. . . My hip looked like mashed potatoes. I used a bedpan for 10 1/2 months. Sometimes I had to be cleaned like I was a baby.' Patient presenting personal patient experience, 2012 Metal-on-Metal Hips Meeting [50]

> 'We are the meshies, women whose lives have been irreparably damaged by synthetic transvaginal mesh. . . we were left permanently injured, robbed of our pre-mesh lives, as we knew them. . .. Because the FDA's 510(k) clearance process does not require premarket testing, we went into our mesh implantation surgeries uninformed. We, the injured, became your guinea pigs. . ..' Advocate presenting personal patient experience, 2011 Vaginal Mesh Meeting [51]

In addition to patient and advocate input, physicians called upon their clinical experiences to share patient's journeys with medical device adverse events, and they too sometimes infused their testimonies with a sense of indignation:

> 'Regardless, this inflammation goes completely out of whack, as you have here.

> Women are presenting just like you would see in a rheumatologist's office, hair loss, rashes, joint pain, tired, all of these diffused kind of, well, we really can't pin it down and it doesn't happen all the time. These are classic symptoms of immune symptoms run awry. And these poor, otherwise healthy women are subjected to these horrific, big abdominal repeated operations to try and fix this. But you can't stop this runaway train.' Individual physician presenting clinical experience, 2015 Essure Meeting [52]

> 'I have a been a dentist for 28 years and through the years I have seen how mercury fillings destroy patients' health and lives. . . My time is limited, so I cannot tell you about all the hundreds of patients I have treated who have been damaged by the mercury put in their mouths. However, I am here to show you three patients who are representative of the patients who have had mercury leaching into their bones, gums and teeth and the health problems they have had because of it.' Individual physician presenting clinical experience, 2010 Dental Amalgam Meeting [53]

Speakers leveraging scientific data sources tended to speak in more detached, 'professional' tones. While patients and advocates spoke with authority about patient embodied experiences, researchers, physicians of both types, industry, and the FDA spoke with authority when drawing on data as evidence, sharing informed assumptions and alternative data sources:

> 'And then, based on a review of literature, trying to determine what is the proportion of amalgam versus other materials in people's mouths, what would be an appropriate assumption in that regard? So we said, well, perhaps 50 percent. So I'm going to talk really about comparing that scenario.' Researcher presenting literature review, 2010 Dental Amalgam Meeting [53]

**Table 5. Types of recommendations by speaker type (N = 789)[1].**

| | Advocate (N = 86) | FDA Rep (N = 163) | Industry (N = 235) | Patient (N = 129) | Physician, Professional Organization (N = 81) | Physician, Solo (N = 44) | Researcher (N = 51) |
|---|---|---|---|---|---|---|---|
| No changes*** | 1% | 0% | 5% | 1% | 12% | 14% | 10% |
| Informed consent*** | 10% | 0% | 1% | 12% | 14% | 2% | 0% |
| Labeling/Communication/ Education*** | 24% | 3% | 6% | 11% | 6% | 9% | 4% |
| Mandatory registry*** | 7% | 2% | 3% | 3% | 17% | 4% | 0% |
| Physician training*** | 12% | 1% | 8% | 8% | 26% | 14% | 12% |
| Premarket trials* | 4% | 2% | 1% | 4% | 9% | 2% | 0% |
| Recall*** | 27% | 0% | 0% | 30% | 6% | 14% | 6% |
| Reclassify (higher risk level)*** | 10% | 1% | 0% | 2% | 1% | 4% | 2% |
| Post-market studies** | 14% | 6% | 2% | 7% | 9% | 9% | 0% |
| Other recommendation* | 6% | 1% | 0% | 4% | 2% | 2% | 2% |
| No recommendations made*** | 36% | 90% | 82% | 45% | 48% | 41% | 72% |

* p < .05

** p < .01

*** p < .001

[1] Note: Speakers often advanced more than one recommendation.

'In a nutshell, we've had 649, now, BHRs [hip resurfacing devices] implanted to date. Three hundred patients have greater than three years of follow-up, with an average of 4.4 years of follow-up. We've had a 99.07% survivorship.' Industry presenting own data, 2012 Metal-on-Metal Hips Meeting [54]

At the same time, advocates asserted their authority in presenting their own data and critiquing shortcomings in existing data sources:

'I'm going to talk a little about the data. . . I want to talk about the rupture rate because it's quite misleading. The companies like to talk about rupture rate as a rate of rupture per implant, but in the past, they've also talked about per patient. . . So when you look at the data in the materials that you were given by the FDA and that the FDA has on their website, please keep in mind that it's per implant.' Advocate presenting clinical trials data, 2011 Breast Implants Meeting [55]

Turning to speaker recommendations, there were statistically significant differences in the following types of recommendations: no recommended changes, informed consent, labeling/ communication/ education, mandatory registry, physician training, premarket trials, recalls, reclassifying up, post-market studies, and other recommendations (see Table 5). Solo physicians (14%), physicians speaking on behalf of professional organizations (12%), and researchers (10%) were most likely to recommend no changes. Professional organizational physicians (14%), patients (12%), and advocates (10%) were most likely to recommend informed consent. Advocates were by far the most likely to call for improved labeling, communication, or education (24%). Professional organizational physicians were most likely to recommend a mandatory registry of medical devices (17%) or premarket trials (9%). They were also the most likely to mention the need for physician training (26%), followed by solo physicians (14%), and

advocates and researchers (12% each). Patients (30%) and advocates (27%) were by far the most likely to ask the FDA to recall a device, and advocates were most likely to ask the FDA to reclassify to a higher level of risk (10%). Advocates were also most likely to call for post-market studies (14%), or to make other recommendations (6%), such as patient representation on advisory teams, financial penalties for manufacturers not reporting adverse events, and required coverage of patient costs stemming from device monitoring and adverse events.

Advocates and patients asked the FDA to take the most stringent actions, like recall and reclassification, and also placed other demands on the agency and industry:

'First, rescind approval of the Mentor implants. They did not uphold their part of the bargain for the postapproval study. If they cannot competently do the research to ensure their product safety, they should not be putting their products into the bodies of more than 100,000 women every year.' Advocate recommending recall, 2011 Breast Implants Meeting [55]

'We also request that the FDA notify the public and alert the medical community to the risk of metal toxicity with copper IUD use, and provide an overview of the symptoms of heavy metal toxicity in a public statement, a press release, and a public service announcement.' Advocate recommending labeling, communication and education, 2019 Metal Amalgam Meeting [56]

Speakers who asked the FDA to take no regulatory action discussed preserving patient choice and providing physicians with every available tool to address medical needs:

'This is a decision making process that occurs between the patient and surgeon. Many reconstructive surgeons view transvaginal mesh as an important option in their toolbox.' Professional organizational physician recommending no change, 2011 Vaginal Mesh Meeting [52]

'I urge this Panel not to interfere with this relationship. Do not posture yourself between a patient and their treating dentist. Do not deprive our patients of one of their most basic rights, the right to make their own healthcare decisions.' Professional organizational physician recommending no change, 2010 Dental Amalgam Meeting [57]

Individual physicians and those representing professional organizations were most likely to recommend physician training. This strategy would likewise leave the physician's toolkit intact, and would not necessitate additional FDA regulatory action. Such suggestions situate safety concerns with the surgeons rather than the medical devices:

'It's not the training of the students; it's the retraining of some of the dinosaurs that may still be teaching that is the obstacle.' Individual physician recommending physician training, 2010 Dental Amalgam Meeting [57]

Some advocates responded directly to these recommendations of 'no action':

'There is a group of a pelvic organ prolapse surgeons who are outraged; their toolkit has been raided. These surgeons now take the position that their unskilled counterparts do not properly implant synthetic mesh, causing a large number of complications. I strongly feel these doctors should take a closer look at the bad tool. Have they taken into consideration, perhaps, the design is flawed? I guess not, since they are defending the tools in their toolkit

as opposed to the thousands and thousands of injured patients. And let's not forget the dead ones either.' Advocate recommending recall and reclassification, 2011 Vaginal Mesh Meeting [52]

## Discussion

Between 2000 and 2020, nearly half of FDA medical device advisory meetings focused on implantable devices, demonstrating their importance in this era of increasing biomedicalization [58]. While the majority of all meetings were scheduled to address premarket and device classification issues, nearly a quarter focused on post-market safety concerns.

While the scripted structure of the FDA advisory meetings may not have shaped the content of testimonies, it did shape the relative speaking time of various stakeholders participating in regulatory proceedings. Word counts for initial statements were significantly lower for patients, compared with FDA and industry representatives, and researchers. Furthermore, the initial word counts were lower overall in the women's health meetings, even controlling for the larger number of speakers. Past studies document how patients', and particularly women's, experiences can become erased in regulatory spaces, either because experts speak for them or they self-censor personal stories, such that statistics displace their embodied knowledge [30, 31]. We did not observe those dynamics here, given the embodied evidence patients presented; rather, their lesser speaking time reflects the structural scripting of the OPH which portions out participants' speaking time precisely, with FDA staff silencing participants' microphones when their allotted time ends.

FDA panelists actively shaped speaking time through their capacity to invite further speaker engagement through question and answer exchanges, resulting in even larger differences between patient, advocate, and independent physician voices, compared with other speakers. FDA panelists were significantly more likely to call on FDA representatives, followed by researchers and industry, compared with patients. Among all stakeholders, industry gained the most additional speaking time, representing greater opportunities to be heard [39].

While advocates referenced many of the same types of scientific evidence leveraged by the 'experts' in the room, the panel followed up with them minimally. Likewise, patients were not given the same level of opportunity to expound on their positions beyond their introductory statements, compared with researchers, industry and FDA representatives. Thus, we did not observe the 'lay expert' role that has been effectively exerted by patients and advocates in some past health-based social movements [24].

In terms of content, patients and advocates drew on embodied knowledge and made strong moral claims about the personal damage patients experienced as a result of having medical devices placed inside their bodies [19, 59]. Physicians also spoke about patients' experiences, drawing on their direct clinical practice experiences. Advocates broadened their base of evidence, demonstrating some features of evidence-based activism, which draws on a 'multiplicity of forms of knowledge' and recognizes that 'knowledge is no longer a mere resource for grounding political claims; it is the very target of activism' [21]. As with traditional 'experts' in the meetings, they referenced and critiqued published literature and clinical trials and made the most use of post-market evidence among speakers. Patients and advocates together recommended the most stringent regulatory actions of device recall and reclassification, along with other measures to improve device safety, transparency and monitoring.

While physicians reflected patient experiences in their testimonies, they drew primarily on clinical and scientific evidence, the foundations of medicine's authoritative knowledge [60]. Although solo physicians were in favor of stringent FDA action (recall), along with patients and advocates, most solo and organizational physicians favored actions that preserve access to

medical technologies, including physician training, no regulatory changes, and patient registries. This may reflect physicians' hard-won professional autonomy within patient-doctor decision making [34, 35].

The remaining stakeholders drew primarily on scientific evidence. Researchers drew on a variety of sources and were most likely to present their own data and material science. Material science is an increasingly important aspect of medical device design and monitoring, given the possibilities of material degradation due to chemical, biological and mechanical processes in the body [61]. Researchers aligned with physicians in calling for physician training and no regulatory changes most prominently, which would keep medical technologies available. Industry representatives similarly drew on scientific evidence, including data they collected, and some industry physicians drew on personal clinical experience. They most commonly recommended physician training. None recommended recall or reclassification and they made fewer recommendations on average. The result was a bit surprising given the threat of profit loss which others contend can function to align corporate competitors to co-promote their medical technologies [37]. FDA representatives relied on a variety of evidence bases, most significantly literature review and advanced the fewest recommendations of all stakeholders, which might be expected given their positions with the agency.

Putting these findings together, the relative lack of speaking time for patients, reinforced by minimal exchanges with the panelists, may illustrate that experiential knowledge is less valued in regulatory spaces than other evidence that is perhaps considered more objective, or technologically informed [30, 32]. The careful scripting of the regulatory meeting's strict structure prevented patients from using interruption and other techniques that the 'active patient' may sometimes employ to redirect and assert perspectives in an interactive way [60, 62]. According to Hwang, Avorn and Kesselheim, citizen participation 'acts as a check on bureaucratic activity and serves to improve the quality of regulations by grounding agency decisions in the public interest;' however, given power asymmetries and needs to resolve technical questions, health policy ends up 'favoring views held by trained regulators according to their technical expertise and discouraging participation except by sophisticated interest groups' [39]. Researchers, industry and FDA representatives were able to speak with greater authority in these meetings. Their relative space in these regulatory proceedings was borne both by the scripted structure, and how that script was negotiated dynamically by the panel, which had authority to shape the script through time granted and questions asked.

A systematic review by Conklin, Morris and Nolte show that the end result of public participation in health policy making is unclear and deserves more careful analysis as it has become an imperative for 'enhancing the responsiveness of health-care systems' [63]. This study documents the variable extent of, and content of, public stakeholder participation in the context of the FDA's regulation of implantable medical devices, devices that only physicians can remove from patients' bodies when safety concerns arise. Advisory committees and panels can 'foster the representation of. . .unheard voices' [26], voices that the FDA might not hear otherwise in its deliberations about the safety of medical devices.

Avenues for public input to the FDA continue to evolve, which may increase opportunities available to patients and other public stakeholders to provide their perspectives. The FDA hosts Patient Listening Sessions which may be initiated by the Agency or by the public. While these sessions can focus on biologics, devices, and drugs, among the 48 sessions summarized and available on the FDA website from 2018 to 2022, medical devices are mentioned in only 11 of these proceedings, whereas pharmaceuticals are mentioned in all. This may suggest that this is an underutilized avenue for public input on medical devices [64]. In 2015, the FDA proposed the Patient Engagement Advisory Committee; most recently chartered in October 2021 to meet twice annually, this committee provides the FDA with advice on 'complex, scientific

issues related to medical devices, the regulation of devices, and their use by patients' [65]. This is an important development and area for future research on patient and other public stakeholder participation in medical device regulation.

## Conclusions

Medical technology has a firm place in contemporary medicine; biomedicalization offers 'control over' one's body through medical intervention (such as contraception), but also by 'transformation of' one's body, selves, health' [66]. Pharmaceuticals address everyday aches and pains but also enhance capabilities [67–70]. Surgical implantation of vaginal mesh, hip replacements, and other medical devices have received much less critical attention than pharmaceuticals [5], and may result in manufactured risks at varying levels of prevalence [71, 72].

The FDA is charged with regulating this growing collection of biomedical treatments and their risks, a daunting task in the current technological regime. The agency relies on its advisory committees and panels to help navigate these risks. FDA advisory meetings are therefore one of the few places where medical device patients might have a voice in the regulatory process, a place where the FDA states that 'members of the public have an opportunity to. . .share their insight' [73]. Advisory meetings constitute what Anspach calls an 'ecology of knowledge', and, along with other public stakeholder structures for regulatory participation, should be subject to further critical analysis and cross-case comparison [74, 75].

The work presented here lays a foundation for further inquiry, and addresses a recent call for analyses of power relations in medicine, as they apply to medical technologies [76]. As we have illustrated, different knowledge bases seem to receive differing prioritization in domains of unequal power–patients and their advocates and independent physicians receives the lowest amount of speaking time in the FDA's medical device advisory meetings, when some are calling for increased citizen participation. Thus, the embodied knowledge and direct clinical experience they offer may be overshadowed by the scientific data advanced by researchers, industry representatives and FDA representatives. These analyses illustrate how the structure of FDA advisory meetings can create a regulatory scripting, whereby the voices of industry, researchers, and the FDA are heard more than patients and their advocates in this key location of regulatory decision making.

## Author Contributions

**Conceptualization:** Shelley K. White, Valerie Leiter.

**Data curation:** Shelley K. White, Valerie Leiter, Mi H. Le.

**Formal analysis:** Shelley K. White, Valerie Leiter, Mi H. Le, Caitlyn K. Helms.

**Methodology:** Shelley K. White, Valerie Leiter.

**Project administration:** Shelley K. White, Valerie Leiter.

**Supervision:** Shelley K. White, Valerie Leiter.

**Writing – original draft:** Shelley K. White, Valerie Leiter, Mi H. Le, Caitlyn K. Helms.

**Writing – review & editing:** Shelley K. White, Valerie Leiter.

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
