## [Decision Letter · Decision Letter 0]

27 Sep 2022

PONE-D-22-20740Regulatory Scripting: Stakeholder Participation in Food and Drug Administration Medical Device Advisory MeetingsPLOS ONE

Dear Dr. White,

Thank you for submitting your manuscript to PLOS ONE. After careful consideration, we feel that it has merit but does not fully meet PLOS ONE’s publication criteria as it currently stands. Therefore, we invite you to submit a revised version of the manuscript that addresses the points raised during the review process.

We look forward to receiving your revised manuscript.

Kind regards,

Quanzeng Wang

Academic Editor

PLOS ONE

Journal Requirements:

    "Unfunded study"

   "No authors have competing interests"

6. Please include your tables as part of your main manuscript and remove the individual files. Please note that supplementary tables (should remain/ be uploaded) as separate "supporting information" files

Reviewers' comments:

Reviewer's Responses to Questions

**Comments to the Author**

1. Is the manuscript technically sound, and do the data support the conclusions?

Reviewer #1: Yes

Reviewer #2: Partly

2. Has the statistical analysis been performed appropriately and rigorously? 

Reviewer #1: Yes

Reviewer #2: Yes

3. Have the authors made all data underlying the findings in their manuscript fully available?

Reviewer #1: Yes

Reviewer #2: Yes

4. Is the manuscript presented in an intelligible fashion and written in standard English?

Reviewer #1: Yes

Reviewer #2: Yes

5. Review Comments to the Author

Reviewer #1: Thank you very much for the interesting manuscript. I would suggest you consider deleting the table describing the number of panel meetings of each panel as this appears peripheral to your main points. More importantly, your most interesting findings are your nuanced analysis of the role of patients and non-physician advocates and the type and tenor of comments they are likely to make. You may wish to increase your focus on this area and summarize your findings at the end of your manuscript. There is a lot of interesting information in this manuscript, but it may be more powerful if it is slightly more focused on the qualitative analyses that you have performed so nicely.

Reviewer #2: This manuscript provides quantitative and qualitative evidence on stakeholder participation in FDA medical device advisory committee meetings. The study is motivated by the fact that there have been ongoing calls to increase patient/public participation in the health policy making process. Because advisory committees are one forum where a broad range of stakeholders can share views with the FDA (via its external advisory committees) they offer, as the authors put it, “one of the few places where patients might have a voice in the regulatory process.”

I share the authors’ view that advisory committees are an interesting site in which to study public engagement in health policy making. I also believe that the technical aspects of this study—sample selection, statistical analysis, qualitative analysis—are well done and the results are clearly presented. However, I think the authors make a questionable choice with respect to a key aspect of the study design that renders the quantitative findings difficult to interpret.

As the authors rightly note (page 6) the open public comment period makes up only a small part of the advisory committee meeting. Thus, it’s unsurprising that comments of public speakers (patients, advocates, etc.) take up a relatively small part of the overall speaking time at the meeting. The authors do note that committee members, and chairs in particular, have some discretion to alter speaking times through follow up questions. But a priori, it seems unlikely that questions would be so extensive as to shift the balance of which groups speak most at meetings, and indeed, the findings bear this out. On their own, then, I don’t think the quantitative findings are particularly compelling. They suggest that speaking times reflect the established structure of advisory committee meetings. Nonetheless the findings may be worth consideration for publication in in this journal given its aims.

But I think there is a larger challenge in the way the authors interpret the quantitative findings to draw broader conclusions about the weight or value given to patient voice in regulatory decision making. Some representative passages:

“Thus, in addition to an asymmetry whereby industry may have a stronger motivation to participate in policy-making than other stakeholders, the structure seems to reinforce this asymmetry, providing industry greater opportunity than some other parties.”

“While this study did not focus on the impacts of public participation, it documents the variable extent of, and content of, constituent participation – at least one important aspect of public participation. Morone and Kilbreth warn of the potential danger of advisory processes simply legitimating official decisions, without creating true participation.”

“FDA advisory meetings are therefore important loci, one of the few places where patients might have a voice in the regulatory process, though as we have demonstrated, they enjoy a limited hearing based on the current script. Advisory meetings constitute what Anspach calls an ‘ecology of knowledge’, and should be subject to further critical analysis and cross-case comparison.”

I think there are two issues with the interpretation suggested by these and other passages, namely, that the discrepancies in speaking time suggest a lack of regard for patient voice. First, while advisory committee meetings do contain open public hearings, their primary function is not as a forum for public engagement. According to FDA, the “primary role” of the FDA advisory committees is to “provide independent expert advice to the Agency in its evaluation of these regulated products and help the agency move toward making sound decisions based upon reasonable application of sound scientific principles.” (https://www.fda.gov/patients/about-office-patient-affairs/learn-about-fda-advisory-committees). It thus seems justifiable that a larger part of the meeting is given over to representatives of the sponsoring company presenting their data and FDA representative presenting the advisory committee with their interpretation of the data. To suggest that the relatively small amount of time given over to public comments is a problem, the authors should do more to explain why it is plausible to think that the public should play a larger role in what are often quite technical discussions.

Second, advisory committee meetings are by no means the only opportunity for patient engagement at the FDA (https://www.fda.gov/patients/learn-about-fda-patient-engagement). Indeed, as the authors note, given the extent to which patients who speak at advisory committee meetings often have financial ties to sponsoring companies, it is not clear that they are a particularly good forums for anything approaching representative public input on FDA decision making. Other initiatives, like the FDA’s more rigorous program for incorporating patient experience data in regulatory decision making (https://www.fda.gov/drugs/development-approval-process-drugs/assessment-use-patient-experience-data-regulatory-decision-making) seem a far better way to incorporate patient voice. Overall, then, I find the authors’ interpretation of their quantitative findings to be somewhat misleading.

Though I recognize that it would entail a significant revision of the paper, I might suggest dropping the quantitative findings and focusing instead on the qualitative findings.

6. PLOS authors have the option to publish the peer review history of their article (what does this mean?). If published, this will include your full peer review and any attached files.

Reviewer #1: No

Reviewer #2: No

---

## [Author Response · Author response to Decision Letter 0]

20 Dec 2022

As directed in the decision email, we have uploaded our Response to Reviewers as a document with our resubmission. Thank you.

---

## [Decision Letter · Decision Letter 1]

1 Feb 2023

Regulatory Scripting: Stakeholder Participation in Food and Drug Administration Medical Device Advisory Meetings

PONE-D-22-20740R1

Dear Dr. White,

We’re pleased to inform you that your manuscript has been judged scientifically suitable for publication and will be formally accepted for publication once it meets all outstanding technical requirements.

Kind regards,

Quanzeng Wang

Academic Editor

PLOS ONE

Additional Editor Comments (optional):

Reviewers' comments:

Reviewer's Responses to Questions

**Comments to the Author**

1. If the authors have adequately addressed your comments raised in a previous round of review and you feel that this manuscript is now acceptable for publication, you may indicate that here to bypass the “Comments to the Author” section, enter your conflict of interest statement in the “Confidential to Editor” section, and submit your "Accept" recommendation.

Reviewer #2: All comments have been addressed

2. Is the manuscript technically sound, and do the data support the conclusions?

Reviewer #2: Yes

3. Has the statistical analysis been performed appropriately and rigorously? 

Reviewer #2: Yes

4. Have the authors made all data underlying the findings in their manuscript fully available?

Reviewer #2: Yes

5. Is the manuscript presented in an intelligible fashion and written in standard English?

Reviewer #2: Yes

6. Review Comments to the Author

Reviewer #2: I appreciate the authors' thoughtful response to the comments. I believe their revisions have added important context for the findings and struck a more effective balance between the quantitative and qualitative sections of the paper.

7. PLOS authors have the option to publish the peer review history of their article (what does this mean?). If published, this will include your full peer review and any attached files.

Reviewer #2: No

---

## [Editor Report · Acceptance letter]

8 Feb 2023

PONE-D-22-20740R1 

Regulatory Scripting: Stakeholder Participation in Food and Drug Administration Medical Device Advisory Meetings 

Dear Dr. White:

I'm pleased to inform you that your manuscript has been deemed suitable for publication in PLOS ONE. Congratulations! Your manuscript is now with our production department. 

Kind regards, 

on behalf of

Dr. Quanzeng Wang 

Academic Editor

PLOS ONE